# High-Speed Optical Chaotic Data Selection Logic Operations with the Performance of Error Detection and Correction

**Geliang Xu** [1,*]**, Kun Wang** [2]**, Liang Xu** [1] **and Jiaqi Deng** [1]

1   School of Electronic Engineering, Chaohu University, Hefei 238024, China; 22003090@chu.edu.cn (L.X.); 21003056@chu.edu.cn (J.D.)
2   School of Computer Science and Artificial Intelligence, Chaohu University, Hefei 238024, China; 22050047@chu.edu.cn
*   Correspondence: xugeliang1027@163.com

**Abstract:** Based on the chaotic polarization system of optically injected cascaded vertical-cavity surface-emitting lasers (VCSELs), we propose a novel implementation scheme for high-speed optical chaotic data selection logic operations. Under the condition where the slave VCSEL (S-VCSEL) outputs a chaotic laser signal, we calculate the range of the applied electric field and the optical injection amplitude. We also investigate the evolution of the correlation characteristics between the polarized light output from the periodic poled LiNbO3 (PPLN) and the S-VCSEL as a function of the optical injection amplitude under different applied electric fields. Furthermore, we analyze the polarization bistability of the polarized light from the PPLN and S-VCSEL. Based on these results, we modulate the optical injection amplitude as the logic input and the applied electric field as the control logic signal. Using a mean comparison mechanism, we demodulate the polarized light from the PPLN and S-VCSEL to obtain two identical logic outputs, achieving optical chaotic data selection logic operations with an operation speed of approximately 114 Gb/s. Finally, we investigate the influence of noise on the logic outputs and find that both logic outputs do not show any error symbols under the noise strength as high as 180 dBw. The anti-noise performance of logic output $O_1$ is superior to that of optical chaotic logic output $O_2$. For noise strengths up to 185 dBw, error symbols in $O_2$ can be detected and corrected by comparison with $O_1$.

**Keywords:** chaotic; data selection; polarization bistability; vertical-cavity surface-emitting laser

## 1. Introduction

As is well known, chaotic laser signals are highly sensitive to system initial conditions and external disturbances, exhibiting a high degree of randomness. Currently, they are widely applied in various fields such as chaotic laser radar, physical random numbers, chaotic neural networks, image recognition and encryption and optical reservoir computing [1–8]. Moreover, they also have potential applications in chaos computing.

Compared with the transmission capacity of optical chaotic networks, the switching capacity of them is still relatively weak. At present, dynamic packet switching technology for optical chaotic signals is an effective measure to enhance the switching capability of optical chaotic networks. The foundation for developing dynamic packet switching technology is to achieve optical chaotic signal processing such as multiplexing, demultiplexing, switching, regeneration and storage involved in optical chaotic signal packet switching nodes. However, the prerequisite for achieving the above chaotic signal processing is to develop low-power, low-loss and high-speed optical chaos computing. Therefore, chaos computing has attracted the great interest of many scholars.

Vertical-cavity surface-emitting lasers (VCSELs), as a microchip semiconductor laser, exhibit many advantages over edge-emitting lasers, in some areas such as low threshold current, single longitudinal mode operation, high modulation frequency, low cost and large-scale integration into two-dimensional arrays [9–11]. When VCSELs are subjected to optical

injection or optical feedback, they easily generate chaotic x-PC and chaotic y-PC which are orthogonal to each other [12–21]. The dynamic behaviors such as polarization switching and polarization bistability in VCSELs can be induced by changing the pump current, the strength of the injected light or the detuning of the injected light [22–26]. Therefore, based on the high-dimensional chaotic system of VCSELs, optical chaotic logic operations with different functions can be designed and implemented by using its rich polarization modes.

In recent years, research in chaos computing has primarily focused on nanosecond-scale basic computations, with very few studies at the picosecond scale. For example, based on a VCSEL with tunable optical injection and polarization bistability, C Masoller et al. implemented an all-optical stochastic logic gate [9–11]. Based on electro-optic (EO) modulation theory and the VCSEL subjected to external optical injection, Zhong et al. achieved optoelectronic composite logic gates such as AND, NAND, OR, NOR, XOR and XNOR [25]. Based on chaos synchronization theory, Yan implemented all-optical logic gates [12]. Our previous works focused on reconfigurable chaotic logic operations [27,28]. The above implementation of chaotic logic gates almost lacks error detection and correction capabilities. However, the development of more complex combinational chaotic logic operations, such as optical chaotic data selection, is still relatively lagging behind, and few people pay attention to it. In 2019, using the polarization bistability in a VCSEL injected by a sampled grating distributed Bragg reflector laser, Zhong et al. first implemented picosecond-level optical chaotic data selection logic operations, and the operation speed was up to 10 Gb/s [29]. In Zhong's scheme, the main reasons limiting the operational speed and noise resistance are twofold: insufficient mutual inhibition between x-PC and y-PC and the poor capability of the threshold demodulation mechanism for short bit durations. In the presence of significant noise, logic outputs may produce errors, but the scheme does not provide appropriate error detection and correction methods. Motivated by these issues, we propose a novel approach for achieving high-speed optical chaotic data selection logic operations with error detection and correction capabilities. In our approach, based on EO modulation theory, we achieve complete mutual inhibition between x-PC and y-PC. By employing an outstanding mean comparison demodulation mechanism, we enable the optical chaotic data selection logic operations to achieve a maximum speed of up to 114 Gb/s. Additionally, leveraging the generalized chaos synchronization theory, the system is capable of accurately detecting and correcting errors in chaotic logic outputs, with a noise strength limit as high as 185 dBw, exceeding the noise strength limit reported in reference [28] by approximately 100 dBw. These research results have great applications and reference values for the construction of high-speed optical chaos computing architecture and the development of optical chaotic network communication systems.

## 2. Theory and Model

Figure 1 depicts a detailed optical path diagram for implementing high-speed optical data selection logic operations. The optical isolator (IS) prevents feedback from the fiber polarization beam splitter (FPBS) into the laser. The fiber beam splitter (FBS) divides light into multiple beams. Meanwhile, the neutral density filter (NDF) controls light intensity. The Faraday rotator (FR) and half-wave plate (HWP) change the polarization direction of light. The optical amplifier (OA) boosts light power. Since the distributed feedback (DFB) laser can emit polarized light in any direction, to ensure that arbitrary polarized light from $IS_1$ can be parallelly injected into the x-PC and y-PC of the master VCSEL (M-VCSEL), it is necessary to convert the arbitrary polarized light into linearly polarized light by some optical passive devices placed between the $IS_1$ and the fiber coupler ($FC_1$). Here, we consider that the converted linear polarization light is the x-PC. Arbitrarily polarized light from the DFB laser is split into x-PC and y-PC by the $FPBS_1$. The y-PC is converted into x-PC by $FR_1$ and $HWP_1$, and then these two x-PCs are coupled into a single x-PC beam by the $FC_1$. Arbitrarily polarized light emitted from the DFB laser is initially split into x-PC and y-PC, using the $FPBS_1$. The y-PC is converted into x-PC through the use of $FR_1$ and $HWP_1$. Subsequently, these two x-PC beams are combined into a unified x-PC beam using

FC$_1$. The unified x-PC beam is divided into three separate beams using a 1 × 3 FBS$_1$. The amplitudes of these beams are individually denoted as $E_{inj1}$, $E_{inj2}$ and $E_{inj3}$. Each beam's intensity is controlled by NDF$_1$, NDF$_2$ and NDF$_3$, respectively. Here, the amplitudes $E_{inj1}$ and $E_{inj2}$ are modulated as optical logic inputs $I_1$ and $I_2$, respectively, while the amplitude $E_{inj3}$ is modulated as the optical clock signal $I_3$. The three beams of x-PC are combined into a single beam using FC$_2$. This unified x-PC beam is then split into two separate x-PC beams using a 1 × 2 FBS$_2$. One of these x-PC beams is directed into the M-VCSEL after passing through NDF$_4$. The second x-PC beam is converted into y-PC using FR$_2$ and HWP$_2$, before being injected into the M-VCSEL following passage through NDF$_5$. The polarized light emitted by the M-VCSEL is separated into x-PC and y-PC by the FPBS$_2$ after transmission through the IS2. The x-PC, designated as o-light, is directly introduced into the periodic poled LiNbO3 (PPLN). Conversely, the y-PC, referred to as e-light, is directed into the PPLN after polarization conversion facilitated by FR$_3$ and HWP$_3$. Under the influence of the applied electric field $E_A$, the PPLN exhibits the electro-optic effect. The o-light (x-PC) generated by the PPLN is directed into the S-VCSEL following amplification by OA$_1$. Conversely, the e-light output is converted into y-PC using FR$_4$ and HWP$_4$, before being introduced into the slave VCSEL (S-VCSEL) subsequent to amplification by OA$_2$. The time delay between the light emission from the PPLN to its arrival at the S-VCSEL is denoted as $\tau$. Subsequently, the light emitted by the S-VCSEL undergoes polarization separation into x-PC and y-PC using FPBS$_3$, following its transmission through the IS$_3$. Here, the logic outputs $O_1$ and $O_2$ are derived by demodulating the x$_P$-PC and y$_P$-PC outputs from the PPLN and the x$_S$-PC and y$_S$-PC outputs from the S-VCSEL, respectively. The subscripts P and S denote the origins from the PPLN and S-VCSEL, respectively.

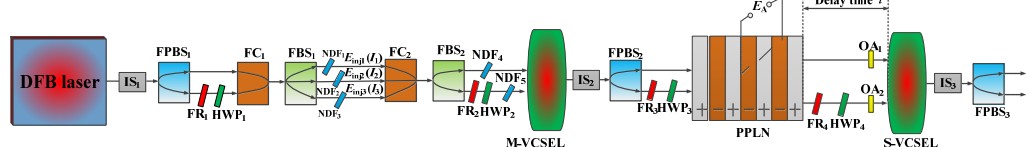

**Figure 1.** Schematic diagram of high-speed optical chaotic data selection logic operations.

In our experimental setup, the applied electric field $E_A$ used to modulate the control logic signal $C$ is a square wave signal characterized by distinct low and high levels, denoted as $E_{A1}$ and $E_{A2}$, respectively, i.e., $C = 0$ when $E_A = E_{A1}$, and $C = 1$ if $E_A = E_{A2}$. Under the appropriate conditions of $E_A$ and utilizing a suitable logic output demodulation mechanism, data selection logic operations can be effectively implemented, i.e., $O_1(t) = I_1(t)\overline{I_3}(t) + I_2(t)I_3(t)$. According to the theory of generalized chaos synchronization, when the S-VCSEL undergoes strong light injection, its output polarization light exhibits high correlation with the polarization light emitted from the PPLN. Therefore, it is feasible to establish $O_2(t + \tau) = O_1(t) = I_1(t)\overline{I_3}(t) + I_2(t)I_3(t)$. This expression signifies the successful implementation of delay storage in data selection logic operations.

Under the influence of spontaneous emission noise, based on the SFM of VCSEL, the rate equation of M-VCSEL subjected to external optical injection can be expressed as follows:

$$\frac{dE_{Mx}(t)}{dt} = k(1 + ia)[N_M E_{Mx}(t) + in_M E_{My}(t) - E_{Mx}(t)] - i(\gamma_p + \Delta\omega)E_{Mx}(t) \\ -\gamma_a E_{Mx}(t) + \sqrt{\beta_{sp}\gamma_e N_M}\xi_x + k_x E_{inj} \tag{1}$$

$$\frac{dE_{My}(t)}{dt} = k(1 + ia)[N_M E_{My}(t) - in_M E_{Mx}(t) - E_{My}(t)] + i(\gamma_p - \Delta\omega)E_{My}(t) \\ +\gamma_a E_{My}(t) + \sqrt{\beta_{sp}\gamma_e N_M}\xi_y + k_y E_{inj} \tag{2}$$

$$\frac{dN_M(t)}{dt} = -\gamma_e[N_M(t)(1 + |E_{Mx}(t)|^2 + |E_{My}(t)|^2)] + \gamma_e \mu_M \\ -i\gamma_e n_M[E_{My}(t)E_{Mx}^* - E_{Mx}(t)E_{My}^*(t)] \tag{3}$$

$$\frac{dn_M(t)}{dt} = \begin{aligned} &-\gamma_s n_M(t) - \gamma_e n_M(t)(|E_{Mx}(t)|^2 + |E_{My}(t)|^2) \\ &-i\gamma_e N_M(t)[E_{My}(t)E^*_{Mx}(t) - E_{Mx}(t)E^*_{My}(t)] \end{aligned} \tag{4}$$

In the above formulas, the subscript $M$ represents M-VCSEL; the subscripts $x$ and $y$ represent x-PC and y-PC, respectively; $N$ is the total carrier concentration, while $n$ is the difference in concentration between carriers with spin-up and spin-down; $E$ is the normalized amplitude, and $E = \sqrt{g/\gamma_e}A$, where g is the differential material gain; $A$ is the slowly varying amplitude; $\gamma_e$ is the nonradiative carrier relaxation rate; $a$ is the linewidth enhancement factor; $k$ is the field decay rate; $\gamma_s$ is the spin-flip relaxation rate; $\gamma_a$ is the linear dichroism; $\gamma_p$ is the linear birefringence; $\mu_M$ is the normalized bias current of the M-VCSEL; $\sqrt{\gamma_e N_M \beta_{sp}}$ denotes the noise term, with $\beta_{sp}$ being the spontaneous emission factor, and noise strength $N_{oi} = 10\lg(\sqrt{\gamma_e N_M \beta_{sp}})^2$. $\xi_x$ and $\xi_y$ are two independent Gaussian white noises with a mean value of 0 and a variance of 1; $k_x$ and $k_y$ are the injection strength for x-PC and y-PC, respectively; $E_{inj} = \sqrt{g/\gamma_e}A_{inj}$ is the sum of $E_{inj1}$, $E_{inj2}$ and $E_{inj3}$, where $A_{inj}$ is the slowly varying amplitude of the injected field; $\Delta\omega = \omega_{inj} - \omega_{ref}$ is the detuning between the frequency of the injected field and the reference frequency of the M-VCSEL; $\omega_{inj}$ is the angular optical frequency of the injected field; the reference optical frequency $\omega_{ref}$ is defined as $(\omega_x + \omega_y)/2$, where $\omega_x = -\gamma_p + a\gamma_a$ and $\omega_y = \gamma_p - a\gamma_a$ are the optical frequency of the x-PC and the y-PC of the free-running VCSEL.

The x-PC and y-PC emitted by the M-VCSEL are injected into the PPLN as the initial input of the o-light and the e-light, respectively, so we have

$$U_{o,e}(0,t) = \sqrt{\frac{\hbar\omega_0 V}{S_A T_L v_c n_1 n_2}} E_{Mx,My}(t) \tag{5}$$

where $U_o$ and $U_e$ are the amplitudes of the o-light and the e light, respectively; $\hbar$ is the Planck constant; $S_A$ is the effective area of the light spot; $V$ is the volume of the active layer of the VCSEL; $v_c$ is the light velocity in a vacuum; $T_L = 2n_g v_c/L_v$ refers to the round trip time in the laser cavity; $L_v$ is the length of the laser cavity; $n_g$ is the effective refractive index of the laser active layer; $\omega_0$ is the central frequency of the M-VCSEL; $n_1$ and $n_2$ are the undisturbed refractive indices of the o-light and the e-light, respectively. With the phase mismatch and the weak second-order nonlinear effect, the analytical solutions of the wave coupling equations of the linear EO effect for the two PCs in the PPLN are written as follows:

$$U_{o,e}(L,t) = \rho_{x,y}(L,t)\exp(i\beta_0 L)\exp[i\phi_{x,y}(L,t)]. \tag{6}$$

where $L$ is the length of the PPLN; $\rho_{x,y}$, $\beta_0$ and $\phi_{x,y}$ are presented in Ref. [30]. The amplitudes of x_P-PC and y_P-PC after undergoing electro-optic modulation in PPLN can be represented as follows:

$$E_{Px,Py}(t-\tau) = \sqrt{\frac{S_A T_L v_c n_{1,2}}{\hbar\omega_0 V}} U_{o,e}(L,t-\tau) \tag{7}$$

The rate equations for the S-VCSEL subjected to strong optical injection with a delay of $\tau$ can be represented as follows:

$$\frac{d}{dt}\begin{pmatrix} E_{Sx}(t) \\ E_{Sy}(t) \end{pmatrix} = \begin{aligned} &k(1+ia)[N_S(t)-1]\begin{pmatrix} E_{Sx}(t) \\ E_{Sy}(t) \end{pmatrix} \pm ik(1+ia)n_S(t)\begin{pmatrix} E_{Sy}(t) \\ E_{Sx}(t) \end{pmatrix} \mp (\gamma_a + i\gamma_p)\begin{pmatrix} E_{Sx}(t) \\ E_{Sy}(t) \end{pmatrix} \\ &+\sqrt{\beta_{sp}\gamma_e N_S}\begin{pmatrix} \xi_x \\ \xi_y \end{pmatrix} + \begin{pmatrix} k_{injx} \\ k_{injy} \end{pmatrix}\begin{pmatrix} E_{Px}(t-\tau) \\ E_{Py}(t-\tau) \end{pmatrix}\exp(-i\omega_0\tau + i\Delta\omega_S t) \end{aligned} \tag{8}$$

$$\frac{dN_S(t)}{dt} = \begin{aligned} &-\gamma_e\Big\{N_S(t) - \mu_S + N_S(t)(|E_{Sx}(t)|^2 + |E_{Sy}(t)|^2) \\ &+in_S(t)[E_{Sy}(t)E^*_{Sx}(t) - E_{Sx}(t)E^*_{Sy}(t)]\Big\} \end{aligned} \tag{9}$$

$$\frac{dn_S(t)}{dt} = -\gamma_s n_S(t) - \gamma_e \Big\{ n_S(t)\big(|E_{Sx}(t)|^2 + |E_{Sy}(t)|^2\big)$$
$$+ iN_S(t)[E_{Sy}(t)E_{Sx}^*(t) - E_{Sx}(t)E_{Sy}^*(t)] \Big\} \tag{10}$$

where the subscript $S$ refers to the S-VCSEL; $\Delta\omega_S$ is the center frequency detuning between the M-VCSEL and the S-VCSEL; $k_{injx}$ and $k_{injy}$ are the injection strength of the $x_P$-PC and $y_P$-PC, respectively; $\mu_S$ is the normalized bias current of the S-VCSEL.

## 3. Results and Discussions

We first calculate Equations (1)–(10) using the fourth-order Runge–Kutta method, employing the parameters listed in Table 1. Since the PPLN cannot generate chaotic polarized light, we investigate the influence of the applied electric field $E_A$ and the optical injection amplitude $E_{inj}$ on the dynamics of $x_S$-PC and $y_S$-PC outputs from the S-VCSEL, illustrated in Figure 2a,b. The figures demonstrate that within the parameter space defined by $E_A$ and $E_{inj}$, $x_S$-PC and $y_S$-PC exhibit various typical dynamic states such as single-period oscillation (P1), two-period oscillation (P2), quasi-periodic oscillation (QP) and chaotic behavior (CO). Our focus here is specifically on the evolution of the chaotic state. From Figure 2a, it can be observed that $x_S$-PC remains in a chaotic state within the regions corresponding to 0–42 kV/mm ($E_A$) and 0.1–10 ($E_{inj}$). As the $E_A$ varies from 42 kV/mm to 100 kV/mm, the chaotic state of $x_S$-PC shows quasi-periodic variations with $E_{inj}$. Specifically, $x_S$-PC exhibits chaotic states when $E_{inj}$ falls within the ranges such as 0.18–0.23, 0.89–0.93, 1.59–1.64, 2.3–2.35 and 3.01–3.06, with a period of approximately 0.7. Figure 2b indicates that $y_S$-PC is in a chaotic state across most regions corresponding to 0–42 kV/mm ($E_A$) and 0.07–10 ($E_{inj}$). As the $E_A$ varies from 42 kV/mm to 58 kV/mm, the chaotic state of $y_S$-PC also exhibits quasi-periodic variations with $E_{inj}$.

**Table 1.** The main system parameters.

| Parameters | Value | Parameters | Value |
|---|---|---|---|
| Linewidth enhancement factor $a$ | 3 | Noise strength $N_{oi}$ | 100 dBw |
| Field decay rate $k$ | 300 ns$^{-1}$ | Polar angle $\theta$ | $\pi/2$ |
| Spin relaxation rate $\gamma_s$ | 50 ns$^{-1}$ | Azimuth $\varphi$ | $\pi/2$ |
| Nonradiative carrier relaxation $\gamma_e$ | 1 ns$^{-1}$ | Crystal temperature F | 293 K |
| Dichroism $\gamma_a$ | 2 ns$^{-1}$ | Poled period of crystal $\Lambda$ | $5.8 \times 10^5$ m$^{-1}$ |
| Birefringence $\gamma_p$ | 60 ns$^{-1}$ | Duty ratio $D$ | 0.5 |
| Delay time $\tau$ | 5 ns | Crystal length $L$ | 15 mm |
| Effective area of light spot $S_A$ | 7.0686 μm$^2$ | Refractive index of o-light $n_1$ | 2.24 |
| Length of laser cavity $L_v$ | 3 μm | Refractive index of e-light $n_2$ | 2.17 |
| Effective refractive index of active layer $n_g$ | 3.6 | Frequency detuning $\Delta\omega_s$ | 0 GHz |
| Volume of active layer V | 21.206 μm$^3$ | Frequency detuning $\Delta\omega$ | 150 GHz |
| Normalized bias current $\mu_M = \mu_S$ | 1.2 | Wavelength of VCSEL $\lambda$ | 1550 nm |
| Optical injection strength $k_x = k_y$ | 10 ns$^{-1}$ | Optical injection strength $k_{injx} = k_{injy}$ | 500 ns$^{-1}$ |
| Threshold current of VCSEL $I_{th}$ | 6.8 mA | Bit duration time $T$ | 10 ps |

Based on the theory of generalized chaos synchronization, when the S-VCSEL experiences intense optical injection from the PPLN, significant correlations may arise between $x_P$-PC and $x_S$-PC, as well as between $y_P$-PC and $y_S$-PC. Here, the correlation coefficient $\rho_{x,y}$ is introduced to quantify the magnitude of the correlation between these pairs:

$$\rho_{x,y} = \frac{\big\langle [I_{Px,Py}(t-\tau) - \langle I_{Px,Py}(t-\tau)\rangle][I_{Sx,Sy}(t) - \langle I_{Sx,Sy}(t)\rangle] \big\rangle}{\Big\{ \big\langle [I_{Px,Py}(t-\tau) - \langle I_{Px,Py}(t-\tau)\rangle]^2 \big\rangle \big\langle [[I_{Sx,Sy}(t) - \langle I_{Sx,Sy}(t)\rangle]^2] \big\rangle \Big\}^{1/2}} \tag{11}$$

where $I_{Px,Py}(t-\tau) = |E_{Px,Py}((t-\tau))|^2$, $I_{Sx,Sy}(t) = |E_{Sx,Sy}(t)|^2$ and the symbol $\langle \rangle$ denotes the time average. The correlation coefficient $\rho_{x,y}$ ranges from 0 to 1, with higher values indicating a stronger correlation between the respective pairs.

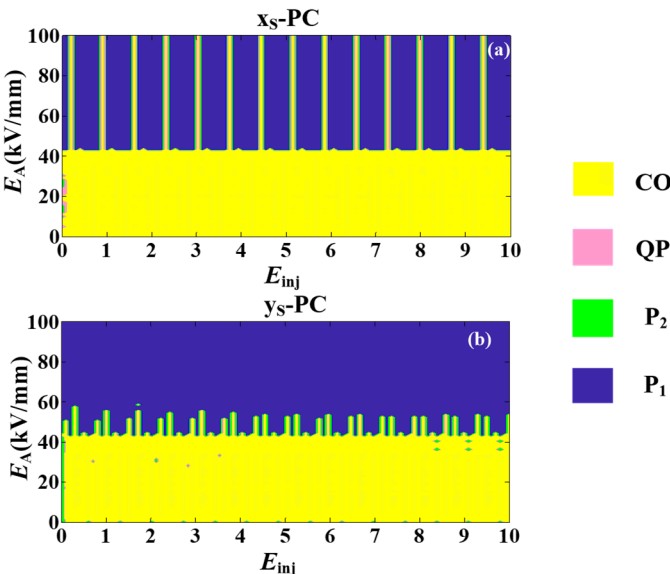

**Figure 2.** Evolutions of dynamic state of $x_S$-PC and $y_S$-PC in parameter space of $E_A$ and $E_{inj}$. (**a**) $x_S$-PC. (**b**) $y_S$-PC.

Here, we calculate the dependence of the correlation coefficient $\rho_{x,y}$ on the optical injection amplitude $E_{inj}$, as shown in Figure 3. Under the condition of $E_A$ = 11 kV/mm as depicted in Figure 3a, as $E_{inj}$ increases from 0 to 2.8, $\rho_x$ gradually rises to 0.73. As $E_{inj}$ further increases to 4.2, $\rho_x$ shows a declining trend, dropping from 0.73 to 0.41. With $E_{inj}$ increasing to 4.4, $\rho_x$ rapidly rises from 0.41 to a peak of 0.97. As $E_{inj}$ progresses from 4.4 to 10, $\rho_x$ slowly decreases to 0.84. For $\rho_y$, as $E_{inj}$ increases from 0 to 3.6, it swiftly rises from 0 to 0.82; with a further increase in $E_{inj}$ to 10, $\rho_y$ decreases slowly to 0.45. From Figure 3b, it is observed that when $E_A$ = 25 kV/mm, the evolution curve of $\rho_x$ with $E_{inj}$ closely matches that in Figure 3a. However, the evolution of $\rho_y$ differs: as $E_{inj}$ increases from 0 to 3.6, $\rho_y$ increases from 0 to 0.82; as $E_{inj}$ continues to 4.2, $\rho_y$ decreases slowly to 0.77; then, as $E_{inj}$ rises from 4.2 to 4.4, $\rho_y$ swiftly increases to a maximum of 0.98. Finally, as $E_{inj}$ progresses to 10, $\rho_y$ decreases gradually from 0.98 to 0.9. In our approach, a stronger correlation is advantageous for enhancing the storage performance of data selection logic operations. Therefore, it is concluded that under the condition where the $E_A$ equals 11 kV/mm, $\rho_x$ reaches a relatively high value around $E_{inj}$ = 4.4, peaking at 0.97. Similarly, $\rho_y$ achieves a relatively high value around $E_{inj}$ = 3.6, reaching a maximum of 0.82. When the $E_A$ is 25 kV/mm and $E_{inj}$ approaches 4.4, both $\rho_x$ and $\rho_y$ reach relatively high values, with $\rho_x$ peaking at 0.96 and $\rho_y$ at 0.98.

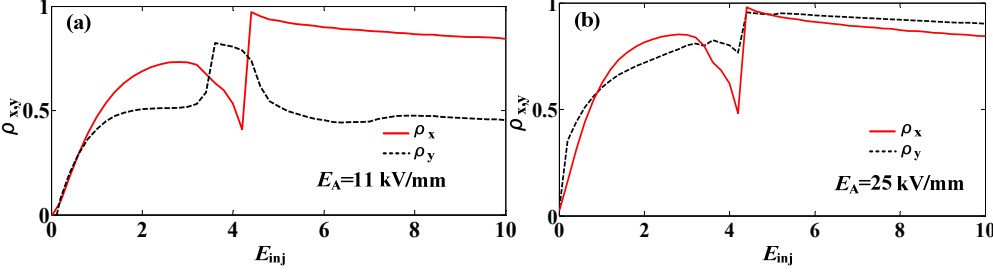

**Figure 3.** The influence of the optical injection amplitude $E_{inj}$ on the correlation coefficient $\rho_{x,y}$ under different applied electric fields. (**a**) $E_A$ = 11 kV/mm. (**b**) $E_A$ = 25 kV/mm.

To further determine the value range of $E_{inj}$ for modulating logic inputs, we calculate the evolutions of the polarization bistability of $x_P$-PC, $y_P$-PC, $x_S$-PC and $y_S$-PC under applied electric fields $E_A$ of 11 kV/mm and 25 kV/mm, respectively, as illustrated in Figure 4. From Figure 4a, it is evident that under an applied electric field $E_A$ of 11 kV/mm,

when $E_{\text{inj}}$ ranges from 0 to 3.9, $x_P$-PC predominates, while $y_P$-PC is suppressed. However, the extent of mutual suppression between them is not pronounced. When $E_{\text{inj}}$ varies between 3.9 and 4.5, both $x_P$-PC and $y_P$-PC display bistable loops, accompanied by an increased degree of mutual suppression. When $E_{\text{inj}}$ ranges from 4.5 to 10, $x_P$-PC dominates completely, while $y_P$-PC is nearly entirely suppressed. In Figure 4b, when $E_{\text{inj}}$ is below 3.9, the evolution of the polarization bistability for $x_S$-PC and $y_S$-PC is similar to that observed in Figure 4a. When $E_{\text{inj}}$ ranges from 3.9 to 4.7, both $x_S$-PC and $y_S$-PC demonstrate bistable loops. As $E_{\text{inj}}$ exceeds 4.7, the mutual suppression between $x_S$-PC and $y_S$-PC intensifies, leading to the complete suppression of $y_S$-PC by $x_S$-PC. From Figure 4c, it is evident that under an applied electric field $E_A$ of 25 kV/mm, when $E_{\text{inj}}$ ranges from 0 to 3.9, $y_P$-PC dominates, while $x_P$-PC is suppressed. When $E_{\text{inj}}$ ranges from 3.9 to 4.7, both $x_P$-PC and $y_P$-PC exhibit bistable loops, with an increased degree of mutual suppression observed between them. When $E_{\text{inj}}$ ranges from 4.7 to 10, $y_P$-PC dominates completely, while $x_P$-PC is entirely suppressed. The polarization bistability evolutions of $x_S$-PC and $y_S$-PC depicted in Figure 4d closely resemble those observed in Figure 4c. In our scheme, the mean comparison mechanism is employed to demodulate $x_P$-PC and $y_P$-PC and $x_S$-PC and $y_S$-PC to obtain logic outputs $O_1$ and $O_2$ (as described below). The bit error rate of $O_1$ and $O_2$ is influenced by the extent of mutual suppression between $x_P$-PC and $y_P$-PC, as well as between $x_S$-PC and $y_S$-PC. A higher degree of mutual suppression correlates with a lower bit error rate. Therefore, based on the above analysis, it can be inferred that the value of $E_{\text{inj}}$ used for modulating the logic input should be greater than 4.7. Considering that maximizing the correlation coefficient $\rho_{x,y}$ is desirable and ensuring that $x_S$-PC and $y_S$-PC maintain a chaotic state, we opt to choose appropriate values of $E_{\text{inj}}$ within the range of 5.35 to 5.46 for modulating the logic inputs.

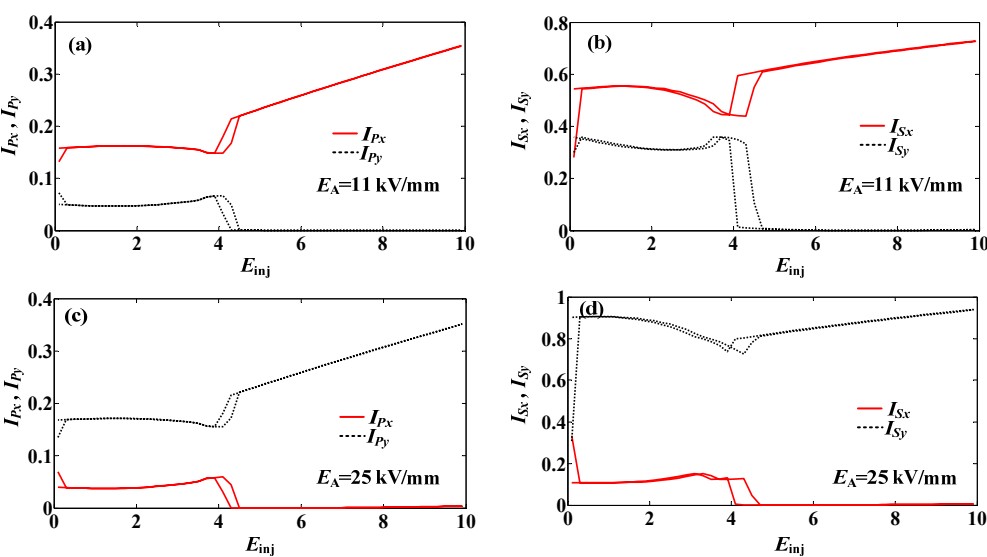

**Figure 4.** The evolutions of the polarization bistability of $x_P$-PC, $y_P$-PC, $x_S$-PC and $y_S$-PC under different applied electric fields $E_A$. (**a**) The polarization bistability of $x_P$-PC and $y_P$-PC under $E_A$ = 11 kV/mm. (**b**) The polarization bistability of $x_S$-PC and $y_S$-PC under $E_A$ = 11 kV/mm. (**c**) The polarization bistability of $x_P$-PC and $y_P$-PC under $E_A$ = 25 kV/mm. (**d**) The polarization bistability of $x_S$-PC and $y_S$-PC under $E_A$ = 25 kV/mm.

Assume that the optical injection amplitude $E_{\text{inj}}$ is equal to the sum of three square wave signals, i.e., $E_{\text{inj}} = E_{\text{inj}1} + E_{\text{inj}2} + E_{\text{inj}3}$, where $E_{\text{inj}1}$ and $E_{\text{inj}2}$ are modulated as optical logic inputs $I_1$ and $I_2$, and $E_{\text{inj}3}$ is modulated as an optical clock signal $I_3$. Since the logic input can be either 0 or 1, there are eight possible input combinations: (0, 0, 0), (0, 0, 1), (0, 1, 0), (0, 1, 1), (1, 0, 0), (1, 0, 1), (1, 1, 0) and (1, 1, 1). Here, we modulate these eight input combinations using four-level signals $E_{\text{injI}}$, $E_{\text{injII}}$, $E_{\text{injIII}}$ and $E_{\text{injIV}}$ (where $E_{\text{injI}}$, $E_{\text{injII}}$, $E_{\text{injIII}}$ and $E_{\text{injIV}}$ are the four possible values of $E_{\text{inj}}$). Specifically, $E_{\text{injI}}$ denotes (0, 0, 0), $E_{\text{injII}}$

represents $(0, 0, 1)$, $(0, 1, 0)$ and $(1, 0, 0)$, $E_{\text{injIII}}$ for $(0, 1, 1)$, $(1, 0, 1)$ and $(1, 1, 0)$ and $E_{\text{injIV}}$ for $(1, 1, 1)$. In this case, the four-level signals remain constant during a bit duration $T$. Here, $T$ is equal to 10 ps, which means the data selection logic operates at a speed of 100 Gb/s. We take $E_{\text{injI}} = 5.37$, $E_{\text{injII}} = 5.4$, $E_{\text{injIII}} = 5.43$ and $E_{\text{injIV}} = 5.46$, which means when $E_{\text{inj1}}$, $E_{\text{inj2}}$ and $E_{\text{inj3}}$ are all equal to 1.79, $I_1$, $I_2$ and $I_3$ are all equal to 0, and if $E_{\text{inj1}}$, $E_{\text{inj2}}$ and $E_{\text{inj3}}$ are all equal to 1.82, $I_1$, $I_2$ and $I_3$ are all equal to 1. Additionally, when $E_A = E_{A1} = 11$ kV/mm, $C = 1$, and if $E_A = E_{A2} = 25$ kV/mm, $C = 0$. The demodulation mechanism for the logic output is described as follows: Suppose that the duration of $C$ being equal to 0 or 1, denoted as $T_1$, is $n'$ times the bit duration $T$, i.e., $T_1 = n'T$. During the time duration $T_1$, the amplitude of $x_P$-PC, $y_P$-PC, $x_S$-PC and $y_S$-PC is sampled $M$ times, where $M$ is equal to $T_1/h$, and $h$ represents the sampling interval. Within $T_1$, the average amplitudes of $x_P$-PC, $y_P$-PC, $x_S$-PC and $y_S$-PC are respectively denoted by $A_{\text{Px}}$, $A_{\text{Py}}$, $A_{\text{Sx}}$ and $A_{\text{Sy}}$:

$$A_{\text{Px}} = \sum_{j=1}^{M} \frac{E_{P_{x(j)}}}{M} \tag{12}$$

$$A_{\text{Py}} = \sum_{j=1}^{M} \frac{E_{P_{y(j)}}}{M} \tag{13}$$

$$A_{\text{Sx}} = \sum_{j=1}^{M} \frac{E_{S_{x(j)}}}{M} \tag{14}$$

$$A_{\text{Sy}} = \sum_{j=1}^{M} \frac{E_{S_{y(j)}}}{M} \tag{15}$$

where $E_{Px(j)}$, $E_{Py(j)}$, $E_{Sx(j)}$ and $E_{Sy(j)}$ respectively represent the $j$th amplitude sampling value of $x_P$-PC, $y_P$-PC, $x_S$-PC and $y_S$-PC within $T_1$. By comparing $A_{\text{Px}}$ and $A_{\text{Py}}$, as well as $A_{\text{Sx}}$ and $A_{\text{Sy}}$, we can determine the logic outputs $O_1$ and $O_2$ within $T_1$. The demodulation rules are as follows: when $A_{\text{Px}} > A_{\text{Py}}$ and $A_{\text{Sx}} > A_{\text{Sy}}$, $O_1 = O_2 = 1$; if $A_{\text{Px}} \leq A_{\text{Py}}$ and $A_{\text{Sx}} \leq A_{\text{Sy}}$, then $O_1 = O_2 = 0$.

We calculate the numerical values of $A_{\text{Px}}$, $A_{\text{Py}}$, $A_{\text{Sx}}$ and $A_{\text{Sy}}$ across different time intervals, as presented in Table 2. Figure 5 illustrates the optical data selection logic operations involving two identical logic outputs. Figure 5a depicts the relationship between the applied electric field $E_A$ and the optical injection amplitude $E_{\text{inj}}$. Based on Figure 5b–e, and Table 2, it is evident that during the time interval from 3 ns to 3.04 ns, where $C = 0$, the logic inputs consist of $(0, 0, 0)$, $(0, 1, 0)$ and $(0, 0, 1)$. In this scenario, $A_{\text{Px}} = 8.74 \cdot 10^{-5} < A_{\text{Py}} = 0.24$, leading to the conclusion that $O_1 = 0$ (as shown in Figure 5f,g); after a delay of $\tau$ ($\tau = 5$ ns), specifically within the time interval from 8 ns to 8.04 ns, it is observed that $A_{\text{Sx}} = 0.0078 < A_{\text{Sy}} = 0.92$. Consequently, we conclude that $O_2 = 0$ (as shown in Figure 5h,i). Therefore, it follows that $O_2(t + \tau) = O_1(t) = I_1(t)\overline{I_3}(t) + I_2(t)I_3(t)$. Between the time interval of 3.04 ns and 3.05 ns, the logic input corresponds to $(1, 0, 0)$ with $C = 1$. Here, $A_{\text{Px}} = 0.25 > A_{\text{Py}} = 5.83 \cdot 10^{-4}$, resulting in $O_1 = 1$. After a delay of $\tau$, specifically within the time interval from 8.04 ns to 8.05 ns, it is observed that $A_{\text{Sx}} = 0.56 > A_{\text{Sy}} = 0.35$. Consequently, $O_2 = 1$. Thus, we establish that $O_2(t + \tau) = O_1(t) = I_1(t)\overline{I_3}(t) + I_2(t)I_3(t)$. Between 3.05 ns and 3.06 ns, the logic input transitions to $(1, 0, 1)$ with $C = 0$. Here, $A_{\text{Px}} = 2.53 \cdot 10^{-5} < A_{\text{Py}} = 0.25$, leading to $O_1 = 0$. Between 8.05 ns and 8.06 ns, it is observed that $A_{\text{Sx}} = 0.31 < A_{\text{Sy}} = 0.75$. Consequently, $O_2 = 0$. Thus, we conclude that $O_2(t + \tau) = O_1(t) = I_1(t)\overline{I_3}(t) + I_2(t)I_3(t)$. The analysis principle for logic operations in other time intervals follows the same methodology; hence, it is not reiterated here. Based on the aforementioned analysis, it is concluded that $O_2(t + \tau) = O_1(t) = I_1(t)\overline{I_3}(t) + I_2(t)I_3(t)$. This signifies the successful implementation of data selection logic operations and their delay storage.

**Table 2.** The numerical values of $A_{\text{Px}}$, $A_{\text{Py}}$, $A_{\text{Sx}}$ and $A_{\text{Sy}}$ in different time periods.

| Time (ns) | $A_{\text{Px}}$ | $A_{\text{Py}}$ | Time (ns) | $A_{\text{Sx}}$ | $A_{\text{Sy}}$ |
|---|---|---|---|---|---|
| 3–3.04 | $8.74 \cdot 10^{-5}$ | 0.24 | 8–8.04 | 0.0078 | 0.92 |
| 3.04–3.05 | 0.25 | $5.83 \cdot 10^{-4}$ | 8.04–8.05 | 0.56 | 0.35 |
| 3.05–3.06 | $2.53 \cdot 10^{-5}$ | 0.25 | 8.05–8.06 | 0.31 | 0.75 |
| 3.06–3.1 | 0.25 | $5.64 \cdot 10^{-4}$ | 8.06–8.1 | 0.5 | 0.19 |
| 3.1–3.11 | $4.01 \cdot 10^{-5}$ | 0.24 | 8.1–8.11 | 0.19 | 0.79 |
| 3.11–3.16 | 0.25 | $5.66 \cdot 10^{-4}$ | 8.11–8.16 | 0.48 | 0.14 |
| 3.16–3.18 | $5.21 \cdot 10^{-5}$ | 0.24 | 8.16–8.18 | 0.11 | 1 |
| 3.18–3.19 | 0.24 | $3.98 \cdot 10^{-4}$ | 8.18–8.19 | 0.5 | 0.33 |
| 3.19–3.2 | $1.02 \cdot 10^{-4}$ | 0.25 | 8.19–8.2 | 0.3 | 0.82 |
| 3.2–3.24 | 0.24 | $5.75 \cdot 10^{-4}$ | 8.2–8.24 | 0.51 | 0.2 |
| 3.24–3.29 | $7.08 \cdot 10^{-5}$ | 0.25 | 8.24–8.29 | 0.04 | 0.9 |
| 3.29–3.3 | 0.25 | $5.83 \cdot 10^{-4}$ | 8.29–8.3 | 0.56 | 0.35 |

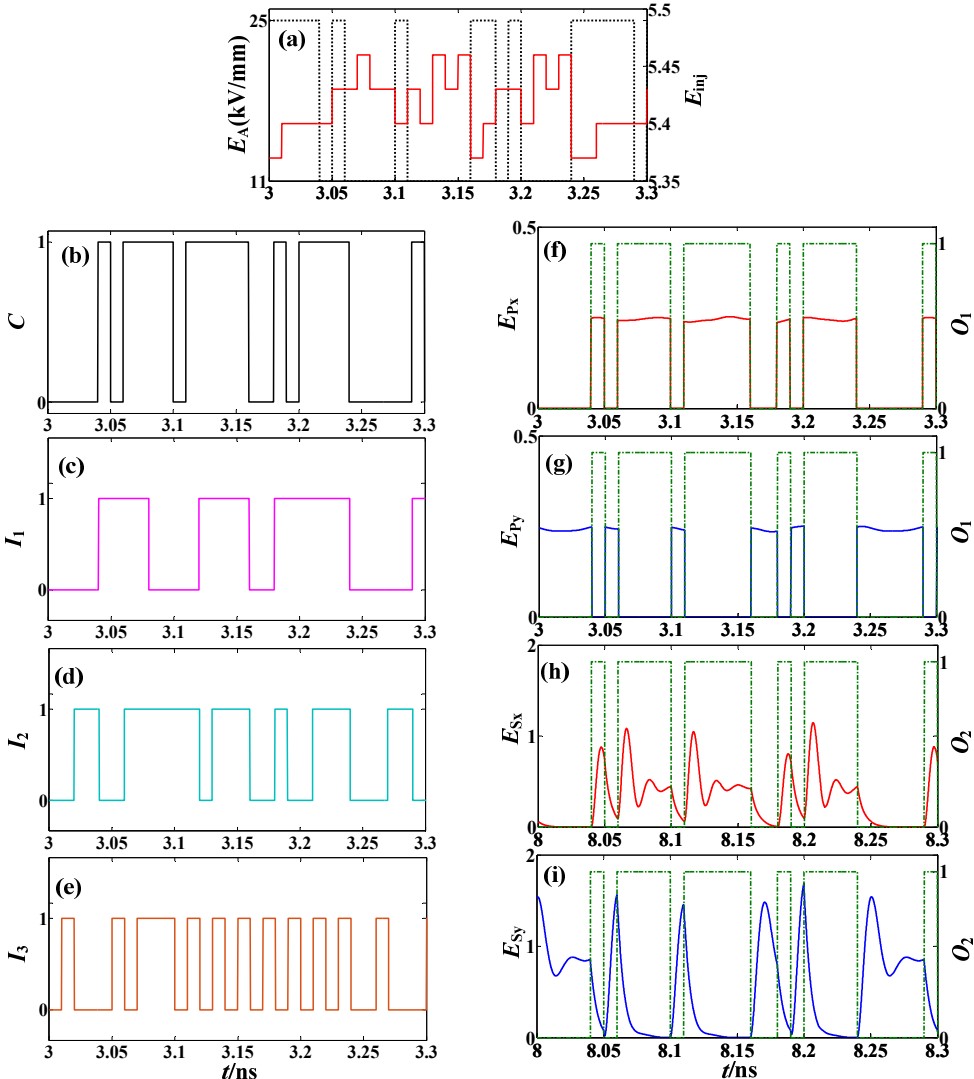

**Figure 5.** The optical chaotic data selection logic operations. (**a**) The combination of $E_{\text{A}}$ and $E_{\text{inj}}$. (**b**) The control logic signal $C$. (**c**) The logic input $I_1$. (**d**) The logic input $I_2$. (**e**) The optical clock signal $I_3$. (**f**) The combination of $E_{\text{Px}}$ and $O_1$. (**g**) The combination of $E_{\text{Py}}$ and $O_1$. (**h**) The combination of $E_{\text{Sx}}$ and $O_2$. (**i**) The combination of $E_{\text{Sy}}$ and $O_2$.

It is crucial to acknowledge that the fidelity of logic outputs is influenced by various system parameters, such as the bit duration time $T$ and the noise strength $N_{oi}$. These factors can introduce bit errors, potentially compromising the integrity of logic operations. Therefore, the success probability (SP) is employed as a metric to quantify the reliability of data selection logic operations. The SP is calculated as the ratio of the number of correctly identified symbols to the total number of symbols present in the output of the logic operations. The evolution of success probability (SP) for the logic outputs $O_1$ and $O_2$ is analyzed across the parameter space defined by noise strength $N_{oi}$ and bit duration time $T$, as depicted in Figure 6. Figure 6a illustrates that for $N_{oi}$ values below 185 dBw, the SP of $O_1$ consistently remains at unity across a $T$ range of 0.25 ps to 50 ps. Beyond 185 dBw, the SP of $O_1$ falls below unity; however, its anti-noise performance can be significantly improved by increasing $T$. Figure 6b reveals that for $T$ values below 8.75 ps, the SP of $O_2$ is less than unity. In the scenario where $N_{oi}$ is less than 180 dBw and $T$ exceeds 8.75 ps, the SP of $O_2$ consistently achieves unity. Nevertheless, as $N_{oi}$ surpasses 180 dBw, the SP of $O_2$ exhibits a gradual decline. From the analysis presented, it can be inferred that to ensure error-free operation for both $O_1$ and $O_2$, the bit duration time $T$ must be set to at least 8.75 ps, and the noise strength $N_{oi}$ should be maintained below 180 dBw. This configuration suggests that the maximum operational speed for optical chaotic data selection is approximately 114 Gb/s. Furthermore, the superior anti-noise performance of $O_1$ compared to $O_2$ is attributed primarily to the more pronounced mutual suppression observed between $x_P$-PC and $y_P$-PC as opposed to that between $x_S$-PC and $y_S$-PC, which is illustrated in Figure 5f–i.

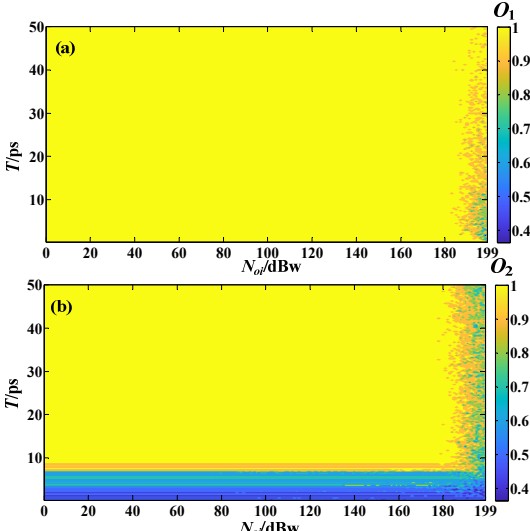

**Figure 6.** The evolutions of SP in the parameter space of $N_{oi}$ and $T$. (**a**) The SP of logic output $O_1$. (**b**) The SP of logic output $O_2$.

In summary, it is evident that when the bit duration time $T$ exceeds 8.75 ps and the noise strength $N_{oi}$ falls within the range of 180 dBw to 185 dBw, the logic output $O_2$ may encounter errors, whereas $O_1$ remains error-free. Consequently, by comparing $O_2$ with $O_1$, it is feasible to detect erroneous symbols within $O_2$ and subsequently implement error correction procedures. In this section, we exemplify the system's error detection and correction capabilities by examining noise strengths of 179 dBw, 185 dBw and 190 dBw. The remaining system parameters are consistent with those depicted in Figure 5. At a noise strength of 179 dBw, Figure 7a illustrates that the noise exerts a pronounced effect on the amplitudes of $x_P$-PC, $y_P$-PC, $x_S$-PC and $y_S$-PC. Nonetheless, leveraging the superior mean comparison demodulation mechanism, the logic outputs $O_1$ and $O_2$ exhibit no errors. At a noise strength of 185 dBw, Figure 7b indicates that the logic output $O_1$ remains error-free. Consequently, by comparing it with $O_1$, it is observed that the logic output $O_2$ incurs errors within the time frame of 8.11 ps to 8.16 ps. These errors are indicated with a black solid

line and marked with the symbol 'x'. The corrected logic outputs are depicted by a pink dotted line. When the noise strength is elevated to 190 dBw, as shown in Figure 7c, the logic output $O_1$ encounters errors within the interval of 3.18 ps to 3.19 ps. Similarly, logic output $O_2$ experiences errors in two distinct intervals: 8.1 ps to 8.11 ps and 8.18 ps to 8.19 ps. Thus, under these conditions, both logic outputs are susceptible to errors, leading to the failure of the logic operations.

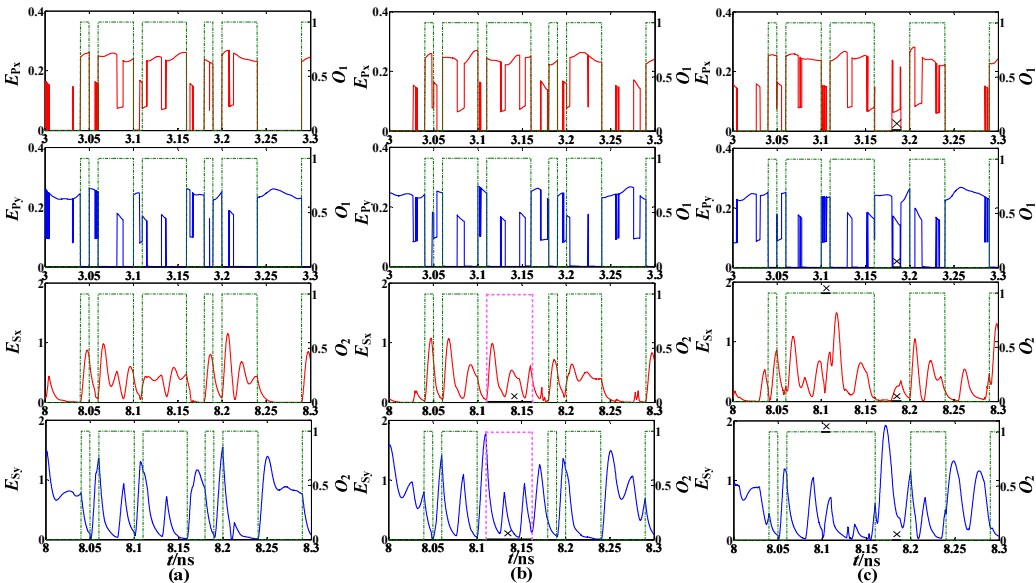

**Figure 7.** The influence of noise strength on logic output $O_1$ and $O_2$. (**a**) $N_{oi}$ = 179 dBw; (**b**) $N_{oi}$ = 185 dBw; (**c**) $N_{oi}$ = 190 dBw.

## 4. Conclusions

Based on the chaotic polarization system of optically injected cascaded VCSELs, we propose a scheme for implementing high-speed optical chaotic data selection logic operations with the performance of error detection and correction. It has been observed that the S-VCSEL can emit chaotic laser output within a broader parameter space encompassing an applied electric field and optical injection amplitude. When the applied electric field is set to 11 kV/mm or 25 kV/mm and the optical injection amplitude is around 4.4, the polarized light emitted by the PPLN exhibits strong correlation with that emitted by the S-VCSEL. Moreover, both the PPLN and S-VCSEL demonstrate identical polarization bistability characteristics: under an applied electric field of 11 kV/mm, x-PC predominates, while y-PC is suppressed. Additionally, y-PC is entirely suppressed by x-PC when the optical injection amplitude exceeds 4.7. Under an applied electric field of 25 kV/mm, the predominant polarization is y-PC, with x-PC being suppressed. Similarly, when the optical injection amplitude exceeds 4.7, x-PC is completely suppressed by y-PC. Based on the aforementioned findings, we utilize the modulation of optical injection amplitude and an applied electric field as the logic input and control logic signal, respectively. Utilizing the outstanding mean comparison mechanism, we achieve two identical logic outputs through demodulating the polarized light from the PPLN and S-VCSEL. This enables the realization of high-speed optical chaotic data selection logic operations, achieving operational speeds of up to approximately 114 Gb/s. Through an examination of noise effects on the logic outputs, it was observed that neither logic output displayed any error symbols even under a noise strength as high as 180 dBw. Furthermore, the anti-noise performance of logic output $O_1$ was found to be superior to that of optical chaotic logic output $O_2$. Error symbols in $O_2$ can be detected and corrected by comparison with $O_1$ when the noise strength does not exceed 185 dBw.

**Author Contributions:** Conceptualization, G.X. and K.W.; methodology, G.X. and K.W.; software, G.X. and K.W.; formal analysis, K.W., L.X. and J.D.; writing—original draft preparation, K.W.; writing—review and editing, G.X.; visualization, K.W. All authors have read and agreed to the published version of the manuscript.

**Funding:** This research was funded by Natural Science Research Project of Anhui Educational Committee (2022AH051724), University-Level General Projects of Chaohu University (XLY-202204), National Undergraduate Innovation and Entrepreneurship Training Program (202310380038).

**Institutional Review Board Statement:** Not applicable.

**Informed Consent Statement:** Not applicable.

**Data Availability Statement:** Data are contained within the article.

**Conflicts of Interest:** The authors declare no conflicts of interest.

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
