# Peer review of "High-Speed Optical Chaotic Data Selection Logic Operations with the Performance of Error Detection and Correction"

_photonics, doi:10.3390/photonics11070586_

Round 1

Reviewer 1 Report

Comments and Suggestions for Authors

In this manuscript (High-Speed Optical Chaotic Data-Selection Logic Operation with the Performance of Error Detection and Correction), authors introduced an implementation scheme to perform high-speed optical chaotic data-selection logic operation based on the chaotic polarization system of optically injected cascaded vertical cavity surface emitting lasers (VCSELs). Authors  calculated the range of the applied electric field and the optical injection amplitude. Furthermore, authors investigated the evolutions of the correlation characteristics between the polarized light output from the periodic poled LiNbo3 (PPLN) and the S-VCSEL as a function of the optical injection amplitude under different applied electric fields. Also, authors calculated the polarization bistability of the polarized light from the PPLN and S-VCSEL.

Authors modulated the optical injection amplitude and the applied electric field as the logic input and the control logic signal, respectively. By using the mean comparison mechanism, authors demodulated the polarized light from the PPLN and S-VCSEL to obtain two identical logic outputs. Authors claimed to achieve optical chaotic data-selection logic operation with the fastest operation speed is approximately 114Gbit/s.

Here is comment on the word presented on this manuscript:

1.     The quality of the written language is relatively poor; I suggest that a native English speaker or a professional language editing service go through the entire manuscript to improve the language quality.

2.     Set keywords in alphabetical order

3.    Introduction Section is poorly written; I suggest that The authors rewrite the section with the following in mind:

a.      Explicitly define the problem leading to the current proposed methodology

b.     I recommend that authors cite the following references:

                 i.      https://doi.org/10.1038/s41598-022-08087-2

               ii.      https://doi.org/10.1117/1.OE.61.7.076104

             iii.      https://doi.org/10.1016/j.jestch.2020.02.007

c.      Clearly define the reason why the proposed detection technique is advantageous when compared with previously reported work.

4.     I recommend that the author explain the main differences between (and the advantage of) the current and (over) the work reported in the following reference:

a.      https://doi.org/10.1364/OE.27.023357

5. Explicity describe the drawback/weakness of the proposed work

6.     While I find the idea of the work presented in this manuscript valuable, the results lack in-depth analysis and discussion. 

Comments on the Quality of English Language

1.     The quality of the written language is relatively poor; I suggest that a native English speaker or a professional language editing service go through the entire manuscript to improve the language quality.

Reviewer 2 Report

Comments and Suggestions for Authors

The article under review is well written on a relevant topic. There are several questions and comments about it:
-in lines 14, 111, it is necessary to correct the spelling of the chemical formula of lithium niobate;
-lithium niobate has a photorefractive effect. Will this scheme work for any wavelength of laser radiation?
-the reviewed article analyzes the nanosecond time scale, and reference 26 talks about the sub-ns time scale;
-what happens in the time interval 3.3–8 ns?

Reviewer 3 Report

Comments and Suggestions for Authors

The authors study the manipulation of the polarization of VCSELs for optical chaotic logic operation.

The paper is easy to read and well written, only a few formal corrections are recommended: in line 33 it should meen "chaos computing"; there should be space before and after mathematical operation signs; there should be space beween numbers and units (sometime there is, sometimes not); in Table 2 I would use dots instead of x - 1.2·10-3 is a scalar poduct not a vector! Some eqations seem to be streched (5-7,9,10...), there should be more space before and after the tables and figures,
